



# 2-D mineral dust radiative forcing calculations from CALIPSO observations over Europe

Maria José Granados-Muñoz[1], Michaël Sicard[1,2], Nikolaos Papagiannopoulos[1,3], Rubén Barragán[1,2], Juan Antonio Bravo-Aranda[4,5] and Doina Nicolae[6]

[1] Remote Sensing Laboratory / CommSensLab, Universitat Politècnica de Catalunya, Barcelona, 08034, Spain

[2] Ciències i Tecnologies de l'Espai - Centre de Recerca de l'Aeronàutica i de l'Espai / Institut d'Estudis Espacials de Catalunya (CTE-CRAE / IEEC), Universitat Politècnica de Catalunya, Barcelona, 08034, Spain

[3] Consiglio Nazionale delle Ricerche, Istituto di Metodologie per l'Analisi Ambientale (CNR-IMAA), Tito Scalo, 85050, Italy

[4] Andalusian Institute for Earth System Research (IISTA-CEAMA), Avd. del Mediterráneo, 18006, Spain

[5] Dpt. Applied Physics, University of Granada, Fuentenueva s/n, 18071, Spain

[6] National Institute of R&D for Optoelectronics (INOE), Magurele, Romania

*Correspondence to*: Maria Jose Granados (maria.jose.granados@tsc.upc.edu)

**Abstract.**

A demonstration study to examine the feasibility to retrieve dust radiative effects based on combined satellite data from MODIS (Moderate Resolution Imaging Spectroradiometer), CERES (Clouds and the Earth's Radiant Energy System) and CALIOP (Cloud-Aerosol Lidar with Orthogonal Polarization) lidar vertical profiles along their orbit is presented. The radiative transfer model GAME (Global Atmospheric Model) is used to estimate the shortwave and longwave dust radiative effects below the CALIPSO (Cloud-Aerosol Lidar and Infrared Pathfinder Satellite) orbit assuming an aerosol parameterization based on CALIOP vertical distribution at a horizontal resolution of 5 km and additional AERONET (Aerosol Robotic Network) data. Two study cases are analysed; a strong long-range transport mineral dust event (AOD = 0.52) originated in the Sahara Desert and reaching the United Kingdom and a weak-





er event (AOD = 0.16) affecting Eastern Europe. The obtained radiative fluxes are first validated in terms of radiative forcing efficiency at a single point with space-time co-located lidar ground-based measurements from EARLINET (European Aerosol Research Lidar Network) stations below the orbit. The methodology is then applied to the full orbit. The obtained results indicate that the radiative effects

5 show a strong dependence on the aerosol load, highlighting the need of accurate AOD measurements for forcing studies, and on the surface albedo. The calculated dust radiative effects and heating rates below the orbits are in good agreement with previous studies of mineral dust, with the forcing efficiency obtained at the surface ranging between -80.3 and -63.0 W·m$^{-2}$ for the weaker event and -119.1 and -79.3 W·m$^{-2}$ for the strong one. Results thus demonstrate the validity of the presented method to retrieve

10 2-D accurate radiative properties with large spatial and temporal coverage.



## 1. Introduction

Mineral dust particles have a strong impact on the atmospheric radiative properties both in the short- and long-wave regions of the radiation spectrum (Sokolik and Toon, 1996; Pérez et al., 2006; Balkanski et al., 2007); affecting also in an indirect way the cloud microphysical properties by acting as cloud condensation and ice nuclei (DeMott et al., 2003; Karydis et al., 2011). The mineral dust effect on the radiation balance of the Earth-atmosphere system is of special relevance due to its large spatial and temporal extent, being one of the most abundant aerosol particles in the atmosphere (Rosenfeld et al., 2001). Its main sources are the arid regions located mostly in northern Africa and western and central Asia. However, it is frequently transported far from its sources to Europe, America and East Asia due to the prevalent global wind patterns (Morman and Plumlee, 2014). The importance of mineral dust in Europe has been recognized and several studies focused on mineral dust vertical distribution making use of ground-based lidar systems (e.g., Ansmann et al., 2003; Mona et al., 2006; Papayannis et al., 2005, 2008; Navas-Guzmán et al., 2013).

There is a need for ground-based stations to coordinate efforts and establish adequate measurement protocols within scientific networks such as the European Aerosol Research Lidar Network (EARLINET; Pappalardo et al., 2014) in order to provide a more regional coverage, which is crucial for the analysis of mineral dust properties from ground (Binietoglou et al., 2015; Granados-Muñoz et al., 2016; Sicard et al., 2016a, 2016b; Barragan et al., 2017). Designated aerosol satellite sensors, even though less accurate, are still a key aspect in order to increase the temporal and spatial coverage for the study of aerosol properties. Sensors such as MODIS (MODerate resolution Imaging Spectroradiometer, Kaufman et al., 2002; Remer et al., 2005) and CALIOP (Cloud-Aerosol Lidar with Orthogonal Polarization (Omar et al., 2010) provide important insight to our understanding of the effects of aerosols on climate. MODIS provides reliable retrievals of spectral aerosol optical depth (AOD) with reasonably well-known uncertainties under most conditions (e.g., Kahn et al., 2010; Levy et al., 2010; Kleidman et al., 2012; Redemann et al., 2006, 2012). Conversely, CALIOP provides vertical profiles of aerosol opti-



cal properties day and night over a narrow swath along the satellite ground track (Winker et al., 2010). The aerosol vertical distribution provided by the CALIPSO mission globally is crucial for aerosol direct radiative effects studies, since radiative properties depend critically on the aerosol vertical distribution (Claquin et al., 1998; Zhu et al., 2007).

In general, aerosol radiative effects are usually based on data collected during intensive measurement campaigns, thus valid mostly locally (Gómez-Amo et al., 2011; Perrone et al., 2012; Meloni et al., 2018; Granados-Muñoz et al., 2019). As a consequence, there is still a gap between the experimental retrieval of the aerosol radiative properties and the modelled estimations used to feed global climate models. The use of satellite measurements such as those performed by CALIPSO provide suitable spa-
tial coverage and temporal resolution that may help to improve the determination of aerosol radiative forcing at regional (Huang et al., 2009; Lemaître et al., 2010; Mallet et al., 2016; Bhawar et al., 2016) or global scales (Oikawa et al., 2013; 2018).

In the present work, we examine the feasibility to retrieve 2-D mineral dust radiative effect along the CALIPSO flight track. Two case studies are presented here, namely a strong event of long-range
transported mineral dust all the way from the Sahara Desert up to the United Kingdom which occurred in April 2011 and a weaker event affecting Eastern Europe on July 2012. A parameterization of the aerosol properties is performed by combining CALIPSO information with ancillary satellite and ground-based data in order to estimate dust radiative properties in both SW (shortwave) and LW (longwave) spectra. The estimated aerosol direct radiative effects obtained with the parameterization just mentioned
are compared and further validated with the aerosol radiative effect calculated at two EARLINET stations located along the CALIPSO track, i.e., Granada (37.2°N, 3.6°W) and Bucharest (44.3°N, 26.0°E). The 1-D radiative effect is estimated using the complete set of aerosol properties necessary for the calculation and well-established methods previously validated in the literature.

The paper is structured as follows: Sect. 2 includes a brief description of the instrumentation and the
datasets used; Sect. 3 is devoted to the description of the GAME radiative transfer model and the meth-



odology used is presented in Sect. 4; Sect. 5 presents the radiative properties of mineral dust obtained and finally, a short summary and concluding remarks are included in Sect. 6.

## 2. Instrumentation and datasets

### 2.1. Satellite-based data

#### 2.1.1. CALIPSO

The CALIOP lidar is the main instrument aboard the CALIPSO satellite. CALIOP measures aerosol backscatter profiles at 532 nm and 1064 nm, including parallel and perpendicular polarized components at 532 nm, at high horizontal and vertical resolution. The high-resolution profiling ability coupled with depolarization measurements make CALIOP an indispensable tool to monitor dust aerosols (Liu et al.,

2008). These measurements are the basis for the Level 2 (L2) data, which includes aerosol and cloud backscatter and extinction coefficients at 532 nm and 1064 nm as well as the particle depolarization ratio at 532 nm (Winker et al., 2009). The generation of the L2 data depends on the successful combination of three modules. First, the processing algorithm separates the atmospheric scene in distinct atmospheric layers (i.e., aerosol, cloud, surface returns; Vaughan et al., 2009). Second, for each aerosol layer

the algorithm classifies the aerosol subtype (i.e., dust, polluted dust, dusty marine, clean continental, polluted continental, marine, and smoke) based on a combination of information, such as the surface type, the layer integrated attenuated backscatter, the depolarization ratio at 532 nm and the aerosol layer height (Kim et al., 2018a). Third, the aerosol backscatter ($\beta_{aer}$) and extinction ($\alpha_{aer}$) coefficient is retrieved assuming lidar ratio values according to the layer subtype (Young et al., 2013). CALIPSO L2

version 4 data are available since 2016 and benefits from significant advances as compared to the predecessor version 3 and older releases (Kim et al., 2018b; Liu et al., 2018; Tackett et al., 2018; Young et al., 2018). In this work we used CALIPSO L2 V4 data, namely the vertical distribution of the extinction coefficient and of the aerosol subtyping at a horizontal resolution of 5 km.



### 2.1.2. MODIS

MODIS is a key instrument on-board the satellites Aqua and Terra that fly as part of the A-Train constellation of satellites, as well as CALIPSO. MODIS measures radiances at 36 wavelengths from 0.41 to 14 μm. Different algorithms are used to retrieve AOD over ocean and over land. These channels have spatial resolutions of 250 m or 500 m and calibration of the radiances is accurate to 2% or better. Radiances are grouped into nominal 10-km cells containing 20×20 pixels at 500-m resolution. The primary sources of uncertainty in MODIS AOD are instrument calibration errors, cloud-masking errors, incorrect assumptions on surface reflectance, and aerosol model selection (Remer et al., 2005; Levy et al., 2010). Retrievals are sensitive to assumptions on surface reflectance, especially over land, where reflectance is higher and more variable than over ocean, and near sun-glint over ocean. These effects become more important as AOD decreases. The AOD retrieval also depends on the fine and coarse mode aerosol models which are used. Selection of an inappropriate model can result in systematic AOD errors. A number of validation studies listed next have characterized uncertainties of the MODIS AOD product. Relative to AERONET AOD measurements, Remer et al., (2005) found that one standard deviation of MODIS-Terra AOD fell within the expected uncertainties of $\tau=\pm0.03\pm0.05\tau$ over ocean and $\tau=\pm0.05\pm0.15\tau$ over land. In this work we use the C6 MYD04_L2 (MODIS/Aqua Aerosol 5-Min L2 Swath 10km) product which contains combined AOD at 550 nm over Land and Ocean. The MODIS level-2 atmospheric aerosol product (MYD04_L2) provides full global coverage of aerosol properties from the Dark Target (DT) and Deep Blue (DB) algorithms. The DT algorithm is applied over ocean and dark land (e.g., vegetation), while the DB algorithm in Collection 6 (C6) covers the entire land areas including both dark and bright surfaces. Results are provided on a 10x10 pixel scale (10 km at nadir). For the spectral surface albedo, the MODIS MCD43C3 Version 6 Albedo Model data set is used (Schaaf and Wang, 2015), providing values at 7 different wavelengths between 0.670 and 2.155 μm. This product is a 5.6 km daily 16-day product, which combines Terra and Aqua data to obtain optimized data of directional hemispherical reflectance (black sky albedo) and bi-hemispherical reflectance (white sky albedo). The MODIS BRDF/ALBEDO products have achieved stage 3 validation.



### 2.1.3. CERES

The CERES (Clouds and the Earth's Radiant Energy System; (Wielicki et al., 1996) instrument, onboard Terra and Aqua satellites alongside MODIS, is a scanning broadband radiometer measuring filtered radiances in the SW (0.3-5μm), total (0.3-200 μm) and LW (8-12 μm) spectral window regions. The Single Scanner Footprint (SSF) product used here contains TOA (Top of the Atmosphere) broadband radiance measurements along with CERES/MODIS-derived cloud and surface information (Wielicki et al., 1998) and provides data with a horizontal resolution of approximately 20km. The integrated emissivity between 4 and 100 μm and the Skin Temperature are SSF Level 2 products and they are used here for the retrieval of the aerosol radiative properties in the LW spectral range.

## 2.2. Ground-based data
### 2.2.1. AERONET Sun-photometer data

AERONET provides world-wide measurements of aerosol optical and microphysical properties. The member stations are equipped with automatic sun-photometers, and more recently, with lunar photometers for night-time operation. Detailed description of the instrument and its function can be found in Holben et al., (1998). The main product of AERONET is the AOD in distinct wavelengths from near ultraviolet (340 nm) to near infrared (1064 nm). The AOD accuracy of calibrated AERONET station is wavelength dependent and varies from root mean square error of ±0.012 (UV band) to root mean square error of ±0.006 (IR band) at overhead sun (airmass=1) (Schmid et al., 1999). AERONET sun-photometers also provide information about aerosol size distribution, aerosol radiative forcing, and aerosol shape. The retrieval of columnar particle size distribution (PSD), asymmetry factor ($g$) and single scattering albedo (SSA), which are used here, is based on the AOD and sky radiance values using an inversion algorithm described in Dubovik and King, (2000) and Dubovik et al., (2006). The uncertainty in the retrieval of SSA is ±0.03 for high aerosol load ($AOD_{440} > 0.4$) and solar zenith angle (SZA) > 50°; while for measurements with low aerosol load ($AOD_{440} < 0.2$), the retrieval accuracy of SSA drops down to 0.02–0.07. For particles in the size range 0.1< r <7 μm (being r the aerosol radius), errors in PSD retrievals are around 10–35%, while for sizes lower than 1 μm and higher than 7 μm retrieval er-



rors rise up to 80–100%. The measurements pass multilevel quality assurance (QA): level 1 without cloud screening, level 1.5 has cloud screening but may be without final calibration. Level 2 has cloud screened and quality assured calibrations.

### 2.2.2. EARLINET lidar data

EARLINET (Pappalardo et al., 2014) operates Raman lidars at a continental scale within the European region. Nowadays, 32 stations are actively providing aerosol extinction and/or backscatter coefficient profiles along with particle depolarization ratio to the EARLINET database, according to EARLINET's measurement schedule (one daytime and two night-time measurements per week). Further measurements are devoted to special events, such as volcanic eruptions, forest fires, and desert dust outbreaks.

The contributing stations have been performing correlative measurements since CALIPSO started its life cycle, based on a schedule established before the satellite mission, and EARLINET has been an important contributor to CALIPSO validation studies (e.g., Mamouri et al., 2009; Mona et al., 2009; Pappalardo et al., 2010; Papagiannopoulos et al., 2016). The standard instruments that the majority of the network operates are multiwavelength Raman lidars, which combine a set of three elastic and two

nitrogen inelastic channels (the so-called 3+2 configuration). This setup allows the independent derivation of the aerosol extinction, $\alpha_{aer}$, at 355 nm and 532 nm, and backscatter coefficients, $\beta_{aer}$, at 355 nm, 532 nm, and 1064 nm during nighttime operation. Besides, the majority of the stations are equipped with depolarization channels, thus providing profiles of the particle linear depolarization ratio. In this study, the EARLINET stations of Bucharest (Romania) and Granada (Spain) are used. The $\beta_{aer}(z,\lambda)$,

being z the vertical height, profiles are obtained from the EARLINET lidar systems during daytime using the Klett-Fernald retrieval (Fernald et al., 1972; Fernald, 1984; Klett, 1981, 1985). The $\alpha_{aer}(z,\lambda)$ profiles are calculated by assuming a height-independent lidar ratio (LR) obtained by forcing the vertical integration of $\alpha_{aer}(z,\lambda)$ to the AOD from AERONET photometer (Landulfo et al., 2003). Uncertainty in the profiles obtained with Klett-Fernald method is usually 20% for $\beta_{aer}(z,\lambda)$ and 25-30% for $\alpha_{aer}(z,\lambda)$

profiles (Franke et al., 2001).



### 2.3. Ancillary data

#### 2.3.1. ERA-Interim

The European Centre for Medium-Range Weather Forecasts (ECMWF) interim reanalysis (ERA-Interim; Dee et al., 2011) is used for the retrieval of the meteorological profiles, namely temperature, pressure, relative humidity and ozone concentration. The dataset used has 0.125°x0.125° horizontal grid spacing with 29 pressure levels from 1000 to 50 hPa. 2-m above ground level (AGL) temperature and surface pressure data are also used. For each CALIPSO pixel along the track, we pick the nearest (by geographical distance) grid point from the ERA-Interim Reanalysis data.

### 3. The GAME radiative transfer model

The dust radiative effects presented in this work are estimated with the GAME radiative transfer model. The GAME code is widely described in Dubuisson et al., (2004, 2005) and Sicard et al., (2014a). It is a modular radiative transfer model that allows calculating upward and downward radiative fluxes at different vertical levels. Particularly, in this study we use 80 vertical levels for the SW (40 for the LW) with decreasing vertical resolution, ranging from 100 m to 1 km, from the surface up to 20 km. The solar and thermal infrared fluxes are calculated in two adjustable spectral ranges, which here are fixed to 297-3100 nm for the SW and 4.5–40 μm for the LW, by using the discrete ordinates method (Stamnes et al., 1988). Note that the GAME code has a variable spectral sampling in the SW (depending on the spectral range considered) and a fixed spectral sampling (115 values) in the LW spectral range. The main input parameters required to feed the model are meteorological profiles, an aerosol model (namely characterized by AOD, SSA and g) and surface parameters such as the surface albedo or the land-surface temperature (LST). In the case of the LW radiative properties calculations, the spectral extinction, SSA and $g$ values are calculated from the particle size distribution using the Mie code. From the radiative fluxes output profiles, the mineral dust radiative effects (DRE) and heating rate (HR) profiles for both the SW and LW components are estimated. More details on the model parameterization can be found in Granados-Muñoz et al., (2019) and detailed information on the properties used for the present study is provided next.



## 4. Methodology

Two different case studies are analyzed in order to demonstrate the feasibility of 2-D mineral dust radiative forcing retrievals along CALIPSO flights under different dust conditions. The first case refers to a strong dust event observed on 7 April 2011 that affected western Europe and was captured by the ground-based EARLINET/AERONET Granada station. The second case refers to a weaker dust event observed over the EARLINET/AERONET Bucharest station on 4 July 2012. Henceforth we refer to the first dust event as GR and the second one as BU. An overview of both events is presented in Figure 1, the MODIS $AOD_{550}$ map illustrates the geographic extent of the dust plume. Besides, the figure shows the flight track of CALIPSO (black solid lines) and the location of the ground-based lidar stations (red squares). The identification of the source of the aerosol layers is made through an analysis of HYSPLIT (Hybrid Single Particle Lagrangian Integrated Trajectory Model; Draxler et al., 1998) back-trajectories. The 5-day HYSPLIT back-trajectories arriving at 2500 m at different points along the CALIPSO track are partly superimposed in the figure and pinpoint the Saharan desert dust as the source of the observed particles.

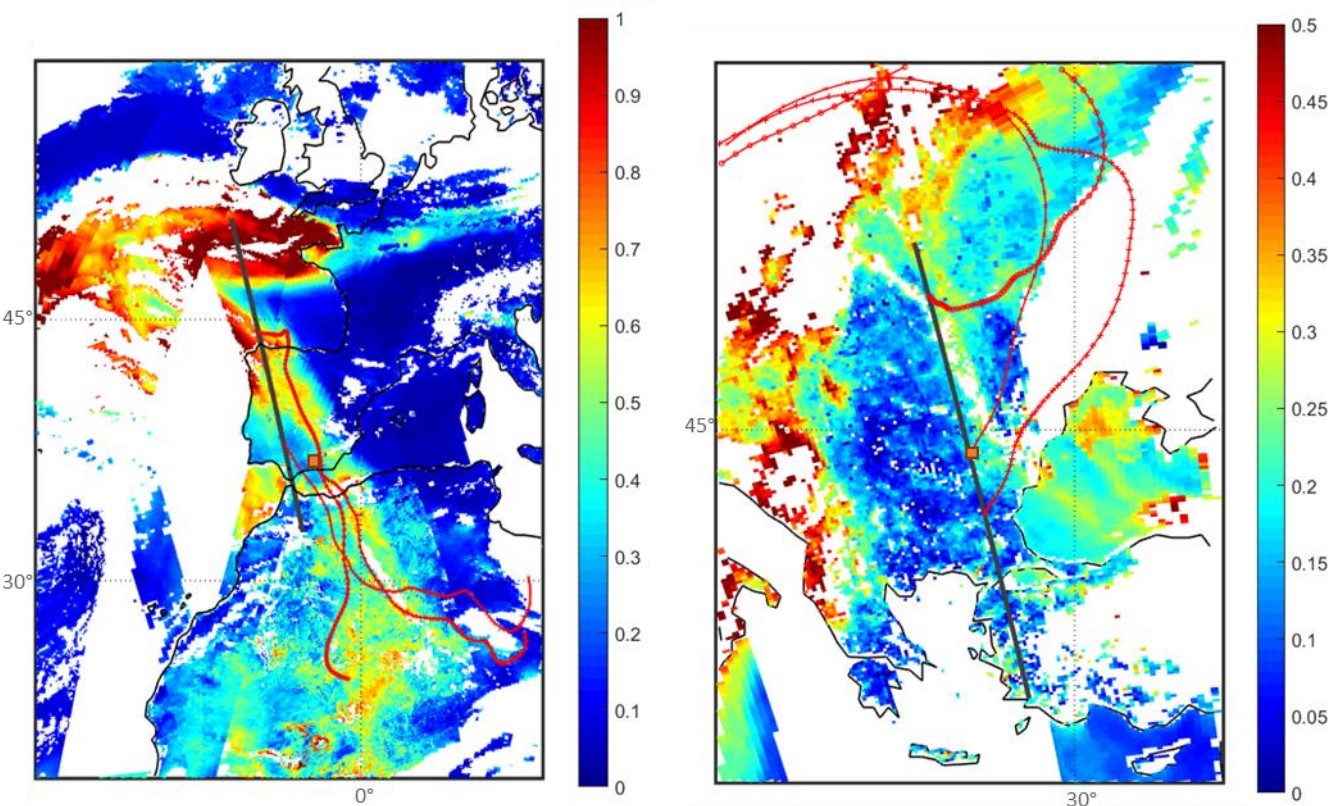

**Figure 1. MODIS AOD values on 7 April 2011 (left) and 4 July 2012 (right). The red squares indicate the position of Granada and Bucharest EARLINET stations and the black lines show the parts of the CALIPSO tracks considered in the current study. Air masses backward trajectories arriving at 2.5km are partly included, indicated by the red dotted lines. Note the different color scales.**

The mineral dust radiative properties profiles are retrieved using GAME below the corresponding CALIPSO orbits. For both cases, the DRE profiles obtained using CALIPSO observations are intercompared with the ones obtained using the ground-based EARLINET lidars when CALIPSO observations and the EARLINET measurements are simultaneous and collocated. For this study, only EARLINET correlative measurements in coincidence with CALIPSO overpasses are used. The term collocated refers to the nearest CALIPSO ground track point to the corresponding ground-based location. The distance between the CALIPSO ground track and the ground-based station is 13.6 km for the Bucharest station and 186.7 km for the Granada station. The latter exceeds the 100 km threshold intro-





duced by Pappalardo et al., (2010) to ensure spatial representativeness between CALIPSO and EAR-LINET, however the severity of the dust event indicated similar dust characteristics for CALIPSO and EARLINET. In the case of Bucharest, the CALIPSO overpass closest to the ground-based station was at 11:28 UT and the lidar measurements are averaged in the interval 11:27-12:27 UT (corresponding to an average solar zenith angle, SZA, of 25.5°). For Granada, CALIPSO overpassed the station at 13:42 UT and the lidar measurements were performed between 13:30 and 14:00 UT (SZA ~ 36.0°).

The DRE estimates at the ground-based stations of Granada and Bucharest with state-of-the-art parametrizations (collocated, simultaneous lidar and sun-photometer retrievals) are performed in order to validate the forcing estimates from CALIPSO at the location of the ground-based stations. For the analysis below the CALIPSO orbit and the analysis at the ground-based sites, two different aerosol input parameterizations are used. The first parameterization (hereafter denoted as PAR1), presented in the current study, uses satellite-based extinction profiles from CALIPSO constrained by MODIS AOD and AERONET SSA, $g$ and PSD in order to retrieve the DRE along the CALIPSO orbit. The total extinction profiles and AOD values considered in the study are assumed to be due to the mineral dust particles, even though some other aerosol types might be present in the mixture. The second parameterization (hereafter denoted as PAR2), based on ground-based EARLINET lidar extinction profiles combined with AERONET data (namely AOD, SSA, $g$ and PSD), has been extensively validated in the literature (Sicard et al., 2014b, 2016a, 2016b; Barragan et al., 2017; Granados-Muñoz et al., 2019) and is used here to retrieve the DRE at the sites of Granada and Bucharest in order to evaluate the results obtained with PAR1.

Meteorological profiles, including vertically-resolved temperature (T), relative humidity (RH) and $O_3$ concentration, are common to both parameterizations and are obtained from the ERA-Interim database. Figure 2 shows the T, RH and $O_3$ profiles obtained along the CALIPSO track for the two case studies presented here, i.e., on 7 April 2011 (GR) and 4 July 2012 (BU). Larger T values and $O_3$ concentration are observed for the BU case in general, even though a stratospheric $O_3$ intrusion is affecting the GR orbit increasing the $O_3$ concentration in some regions. These differences, together with the differences in the solar position, surface and aerosol properties, lead to differences between the radiative





fluxes and the DRE estimated in GR and BU cases. The regions with high RH need to be considered in order to study possible variations in the extinction profiles due to hygroscopic growth or even cloud formation. The surface albedo data are obtained from MODIS for the SW retrievals, whilst CERES LW surface emissivity and skin temperature (or land-surface temperature, hereafter denoted as LST) are used in the retrieval of the LW radiative properties.

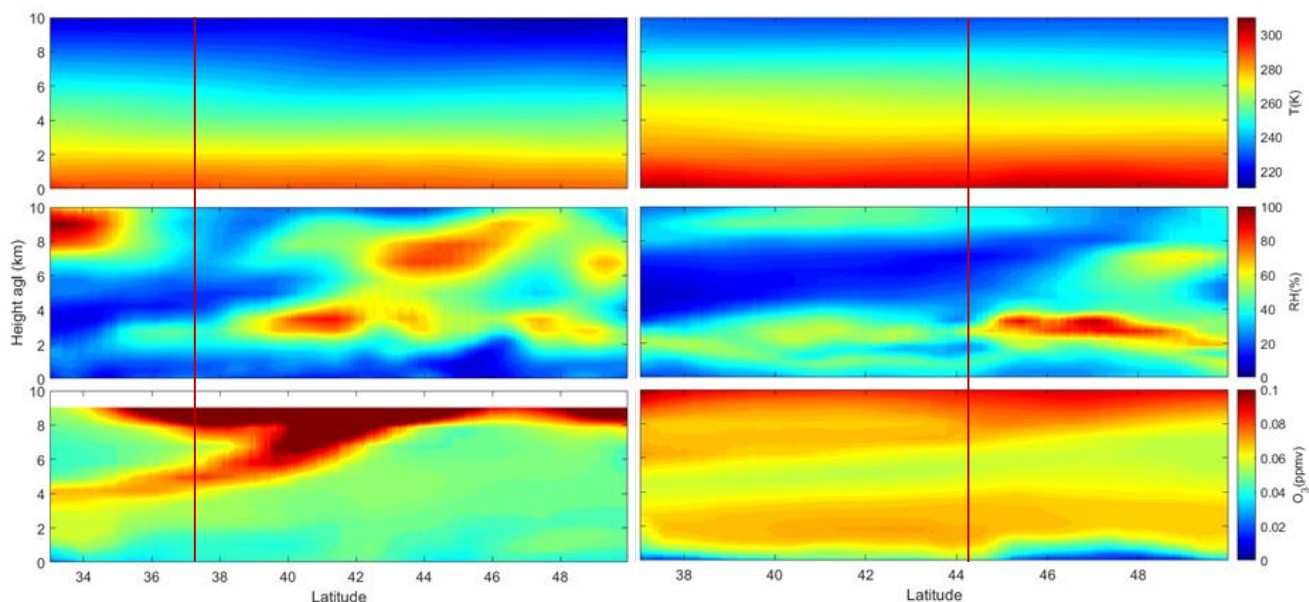

**Figure 2. ERA-Interim temperature (T), relative humidity (RH) and ozone (O₃) profiles (from top to bottom respectively) for 7 April 2011 (GR) and 4 July 2012 (BU). The vertical red lines indicate the position of Granada and Bucharest EARLINET stations.**

### 4.1. Aerosol parameterization using satellite data

Mineral dust radiative properties are obtained along the CALIPSO track for those profiles where mineral dust is detected, according to the aerosol typing provided by CALIPSO. For the parameterization of GAME retrievals along the CALIPSO track, the so-called PAR1, we make use of the $\alpha_{aer}$ profiles provided by CALIPSO, constrained by collocated MODIS AOD values. Although CALIPSO provides AOD information by integrating the $\alpha_{aer}$ profiles, these data are usually affected by large uncertainty (e.g Wandinger et al., 2010; Rogers et al., 2014). Here, we chose to constrain the CALIPSO profiles





with MODIS AOD that has been extensively validated (Levy et al., 2005; Remer et al., 2012; Gupta et al., 2018). Furthermore, the DRE estimations are highly dependent on the AOD, thus using the well-established MODIS data we aim to improve the accuracy of the retrievals. Simultaneous MODIS SW surface albedo data are also used to retrieve the aerosol radiative forcing profiles. Figure 3 shows the

curtain plot of the MODIS constrained $\alpha_{aer}$ profiles obtained along CALIPSO flight track together with the averaged profiles for both GR and BU dust events. MODIS surface albedo values at 675 nm and AOD at 550 nm are also depicted (black and red dots respectively). The $\alpha_{aer}$ is much larger for GR, almost twice than in BU, and the mineral dust layer reaches much higher altitudes (5 km agl, whereas in BU the dust is constrained below 4 km). On the averaged profiles, we can see that the maximum is

found between 1 and 5 km for GR case whereas the profile is quite homogeneous below 4 km in the BU event. In both cases, the AOD is generally increasing with latitude, with larger values and much more variability in GR. For the surface albedo, there is a strong decrease in GR above 44º, where the surface changes from land to ocean (see also Figure 1).

Assuming that the mineral dust optical and microphysical properties are homogeneous for each

one of the events analyzed here, the spectral values of the SSA, *g* and the PSD are taken from the AERONET Level 1.5 inversions at Granada and Bucharest and assumed to be constant along the track. Unfortunately, Level 2.0 data were not available at the time of the measurements. Figure 4 shows the AERONET data used as input in GAME including the SSA (Figure 4a), *g* (Figure 4b) and PSD (Figure 4c). The SSA values and their spectral dependence over Granada and Bucharest are typical of mineral

dust particles, i.e., an increasing SSA with increasing wavelength, even though the large SSA values at 440 nm for Bucharest indicate mixing with anthropogenic pollution from the city in the lower layers near the surface. Although the asymmetry factor has a different behavior at both stations, it is rather spectrally independent (variations lower than 0.04 between 440 and 1020 nm) in both of them which reflects again the typical signature of mineral dust. For the meteorological profiles, data are obtained

from the ERA-Interim Reanalysis (see Fig. 2). LW surface emissivity and LST are provided by CERES.



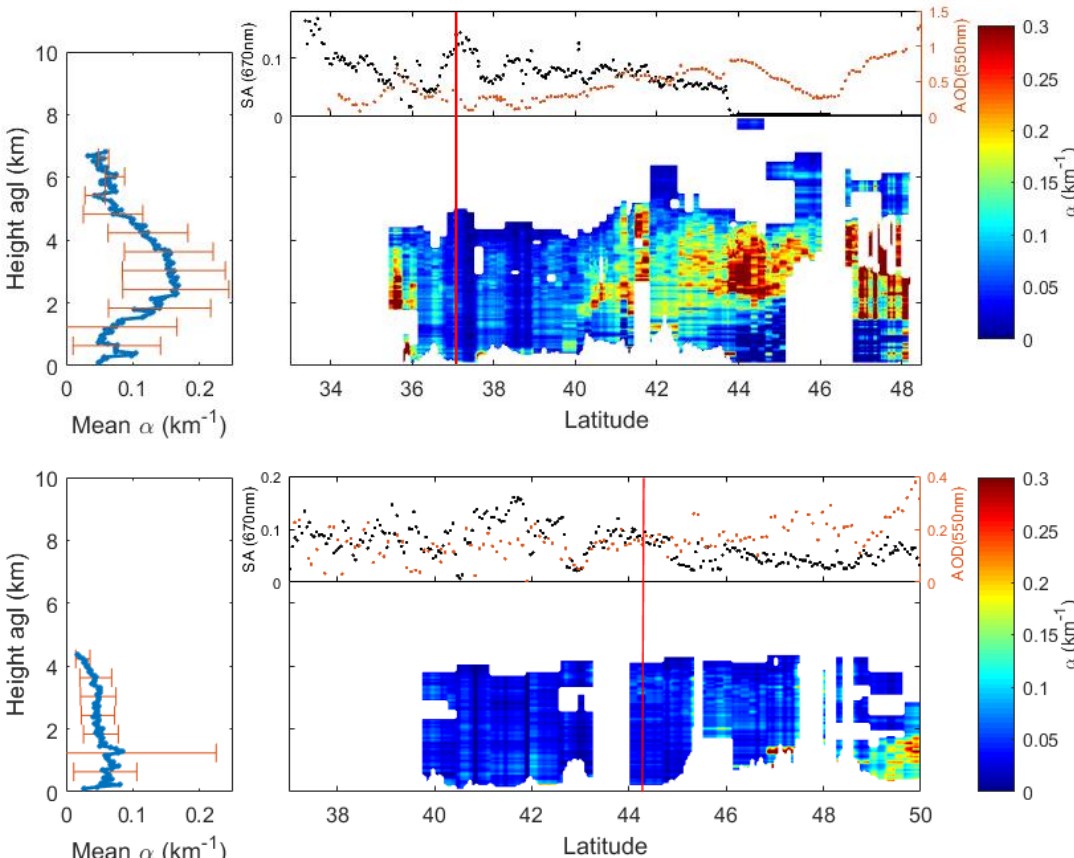

**Figure 3.** Aerosol extinction at 532m as provided by CALIPSO and constrained by MODIS AOD on 7 april 2011 (top) and 4 July 2012 (bottom) and the corresponding averaged profiles for the considered part of the track (left). The error bars are the standard deviation. The red vertical lines represent the location of Granada (top) and Bucharest (bottom) stations. MODIS AOD$_{550nm}$ (orange dots) and surface albedo at 670 nm (black dots) along the track are also depicted in the top part of the curtain plots.





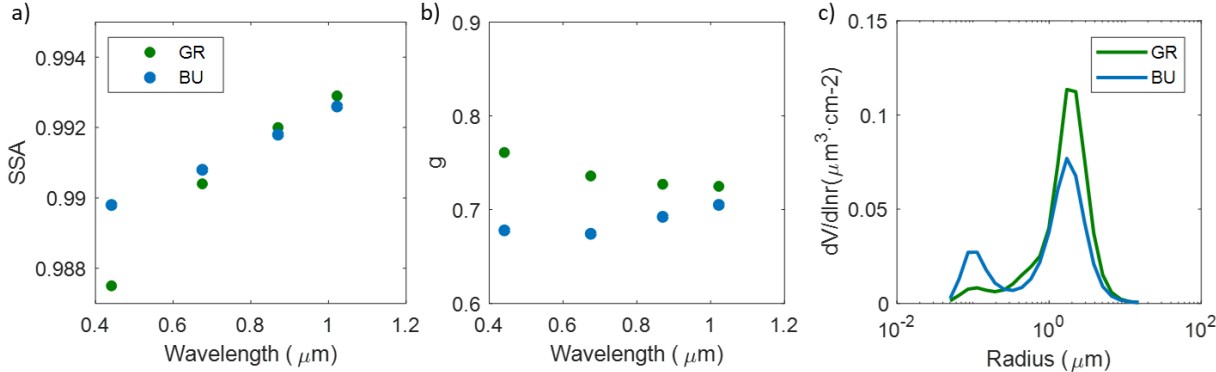

**Figure 4. AERONET Level 1.5 retrieved a) single scattering albedo (SSA), b) assymetry factor (g) and c) particle size distribution for Granada (green) and Bucharest (blue) stations.**

## 4.2. Aerosol parameterization using ground-based data

For PAR2, the $\alpha_{aer}$ profiles containing information about the aerosol vertical distribution are obtained from the ground-based EARLINET lidar systems constrained by the sun-photometer AOD provided by AERONET (Figure 5 and Table 1). The values of the SSA, *g* and the PSD are provided by the AERONET Level 1.5 inversions shown in Fig. 4. The surface albedo at Granada and Bucharest sites is obtained from MODIS data, whereas the longwave emissivity and the LST from CERES are used. Meteorological profiles from ERA-Interim are retrieved for the pixels containing Granada and Bucharest stations. Simultaneous and collocated CALIPSO $\alpha_{aer}$ profiles constrained by MODIS AOD are also depicted in Figure 5. Very similar vertical structures are detected by the ground-based lidar and CALIPSO, even though CALIPSO profiles are expectedly much noisier. As for the $\alpha_{aer}$ values, the satellite profiles slightly underestimate the ground-based measurements. This is directly related to the lower MODIS AOD values compared to AERONET (Table 1). Differences of 0.12 and 0.08 are observed at Granada and Bucharest, respectively, between AERONET and MODIS nearest pixel AODs, which are larger than the combined uncertainty. These differences in the aerosol load are somehow expected due to the distance between the satellite overpasses and the ground-based station, the time difference between the AERONET and MODIS measurements (5 minutes for GR and 20 minutes for BU) and the different horizontal resolution (10 km for MODIS vs a point value measured by AERONET). Since the





DRE is highly dependent on the AOD values, the comparison of the radiative properties at the stations between the retrieval obtained using the EARLINET lidars and the one using CALIPSO is made in terms of the forcing efficiency (FE) to avoid this dependence. The FE is defined here as the ratio between the DRE and the $AOD_{550}$.

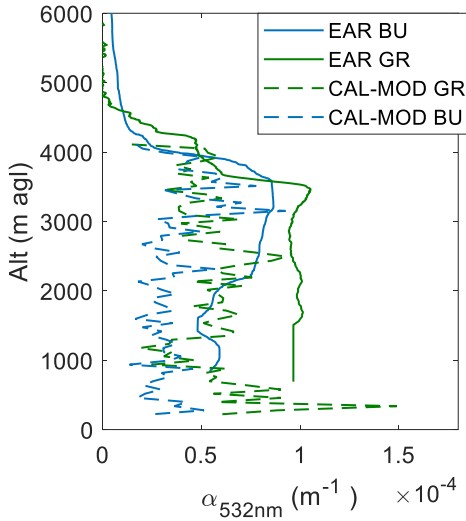

**Figure 5. AOD-constrained aerosol extinction profiles retrieved with the ground-based EARLINET lidars (solid line) and CALIPSO (dashed line) for Granada (green) and Bucharest (blue) stations.**

|        | $AOD_{550}$ (AER/MODIS) | AE(440-870nm) AERONET | SA(470nm) MODIS |
|--------|-------------------------|-----------------------|-----------------|
| **GR** | 0.34/0.22               | 0.27                  | 0.05            |
| **BU** | 0.22/0.14               | 0.75                  | 0.048           |

**Table 1. AERONET and MODIS retrieved AOD at 550 nm, Angstrom exponent between 440 and 870 nm (AE(440-870nm)) retrieved by AERONET and surface albedo (SA) at 470 nm obtained from MODIS data above Granada and Bucharest stations.**

A summary of the data used for the two aerosol input parameterizations (PAR1 and PAR2) can be found in Table 2. The parameters which are variable along the satellite track are indicated in italic bold font.





| | $\alpha_{aer}(z)$ | AOD($\lambda$) | SSA($\lambda$) | g($\lambda$) | SA($\lambda$)(SW/LW) | PSD | Met(z) | LST |
|---|---|---|---|---|---|---|---|---|
| **PAR1** | *CAL* | *MODIS* | | | *MODIS/CERES* | | *ERA-Int* | *CERES* |
| | | | AER | AER | | AER | | |
| **PAR2** | EAR | AER | | | MODIS/CERES | | ERA-Int | CERES |

Table 2. Input parameters used for GAME retrievals of the DRE for parameterizations 1 (PAR1) and 2 (PAR2). Those parameters which are variable along the track are marked in bold italic font. (CAL-CALIPSO; AER-AERONET; EAR-EARLINET; ERA-Int-ERA-Interim)

## 5. Mineral dust radiative properties

### 5.1. Radiative properties profiles at the EARLINET sites of Granada and Bucharest

The aerosol FE profiles obtained with GAME at the EARLINET stations of Granada and Bucharest are presented in Figure 6. The dashed lines represent the profiles retrieved using the PAR1 parameterization based on CALIPSO $\alpha_{aer}$ profiles constrained by MODIS AOD simultaneous and collocated to the EAR-LINET-retrieved $\alpha_{aer}$ profiles used in PAR2, represented here with the solid lines.

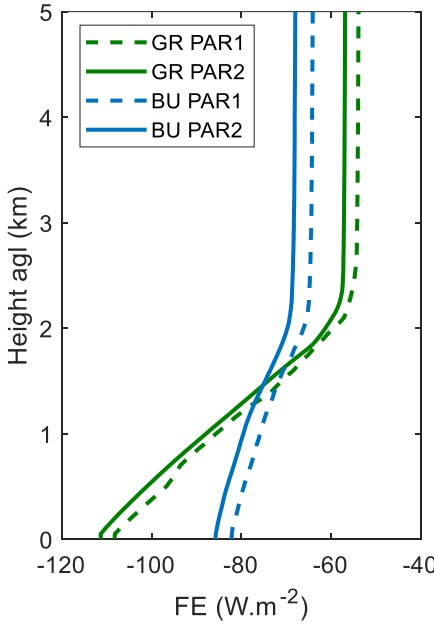

Figure 6. Profiles of the forcing efficiency obtained using CALIPSO (PAR1) and the ground-based EARLINET lidar (PAR2) extinction profiles on 4 April 2011 above Granada (GR) and 7 July 2012 above Bucharest (BU).



In the case of dust outbreak affecting Granada in 2011, we observe that the FE at the surface is much larger (in absolute values), whereas at the TOA lower values are obtained. A similar behavior is observed for Bucharest, even though differences between the FE at the BOA (bottom of the atmosphere) and TOA are smaller. A quite different impact on the FE both at the BOA and the TOA is observed for the two events. Differences in the dust properties and the different SZA affect the radiative impact of the aerosol particles, producing a stronger effect at the BOA for Granada and at the TOA for Bucharest. Altogether, we obtain a cooling effect both at the BOA and the TOA, which is in agreement with previous studies of mineral dust radiative effects (Di Biagio et al., 2009; Gómez-Amo et al., 2011; Meloni et al., 2015). Small differences (lower than 3 Wm$^{-2}$) are obtained between the two parameterizations. Nonetheless, these differences are not significant since they are lower than the uncertainty related to the uncertainties in the input profiles, which can reach 25% according to estimations presented in Granados-Muñoz et al., (2019). Thus, despite the larger noise in the CALIPSO signals compared to EARLINET, the results indicate that the vertical distribution of the mineral dust provided by CALIPSO combined with MODIS AOD provide similar results to that obtained with the ground-based stations. The methodology used here for the ground-based stations has already been extensively validated (Sicard et al., 2014b, 2016a, 2016b; Barragan et al., 2017; Granados-Muñoz et al., 2019) and the extension of this methodology for the CALIPSO space mission allows us to greatly increase spatial and temporal coverage.

### 5.2. DRE below CALIPSO overpass

Figure 7 shows the resulting $DRE_{SW}$, $DRE_{LW}$ and the $DRE_{LW}/DRE_{SW}$ ratio profiles obtained from the GAME simulations for GR and BU dust events. Variations along the satellite tracks are mostly driven by changes in the $\alpha_{aer}$ profiles, with larger absolute values of the DRE observed in regions with higher aerosol load. The high standard deviation and range indicate a large variability of the DRE along the track, with those regions where the AOD is minimum having an almost negligible radiative effect (-8.6 and -0.2 W·m$^{-2}$ at the BOA at GR and BU respectively). Values as large as -169.8 W·m$^{-2}$ at the BOA for the SW are obtained for GR, whereas for BU they reach only -38.0 W·m$^{-2}$. For the LW, the DRE ranges between 1.0 and 20.0 W·m$^{-2}$ at the BOA in GR and between 0.0 and 4.0 W·m$^{-2}$ in BU. The op-



posite effect observed for the LW, where the DRE has a positive sign, counterweights the cooling in the SW range, and averaged net DRE values at the BOA and TOA respectively are -52.9 and -33.5 $W \cdot m^{-2}$ for GR and -12.4 and -10.5 $W \cdot m^{-2}$ for BU, which translates into a decrease of the net DRE ranging between 10 and 20% compared to the SW values. The profiles obtained at the location of the ground-based stations are also included in Fig. 7. In the case of GR, these profiles correspond to a very low aerosol load compared to the rest of the track and the DRE is close to the minimum, whereas for the case of BU is close to the averaged values along the track. For the $DRE_{LW}/DRE_{SW}$ ratio similar values are obtained on average for both cases, with the averaged profiles ranging between 10 and 15% and maximum values always below 25%, indicating similar relative LW-to-SW properties of the two events despite the differences in the optical and microphysical properties observed between the events. For GR, a strong increase in the DRE, especially in the SW, is observed for latitudes above 44°, where the shift from land to ocean occurs. Above this latitude, there is a strong decrease of the SW surface albedo (Fig. 4), whereas for the LW, the surface albedo slightly increases, from 0.013 to 0.018 (not shown). Actually, the average $DRE_{SW}$ ($DRE_{LW}$) at the BOA is -91.0 $W \cdot m^{-2}$ (11.6 $W \cdot m^{-2}$) for the ocean portion and it al almost halved for the land part, being -46.9 $W \cdot m^{-2}$ (7.7 $W \cdot m^{-2}$). The influence of the surface albedo is more important for the SW compared to the LW and, thus, lower $DRE_{LW}/DRE_{SW}$ ratio values are obtained over 44°N.

The strong dependence of the DRE on the $\alpha_{aer}$ values and, thus, on the aerosol load is also evident in the strong negative correlation observed between the DRE at the surface and the AOD. Figure 8 shows the MODIS AOD at 550 nm and the DRE values at the surface along the CALIPSO track for GR and BU. For both cases it is observed a general increase of the $AOD_{550}$ as the latitude increases (see also Figure 1), even though the values for GR are more variable. The increase in $AOD_{550}$ is very well correlated with the increase (in absolute value) of the total DRE at the BOA. Correlation coefficients equal to -0.98 for Granada and -0.70 for Bucharest are obtained. This strong dependence of the DRE on the AOD is already well-documented (Prasad et al., 2007; Sicard et al., 2014a; Lolli et al., 2018; Meloni et al., 2018) and, consequently, using the FE we avoid this dependency.

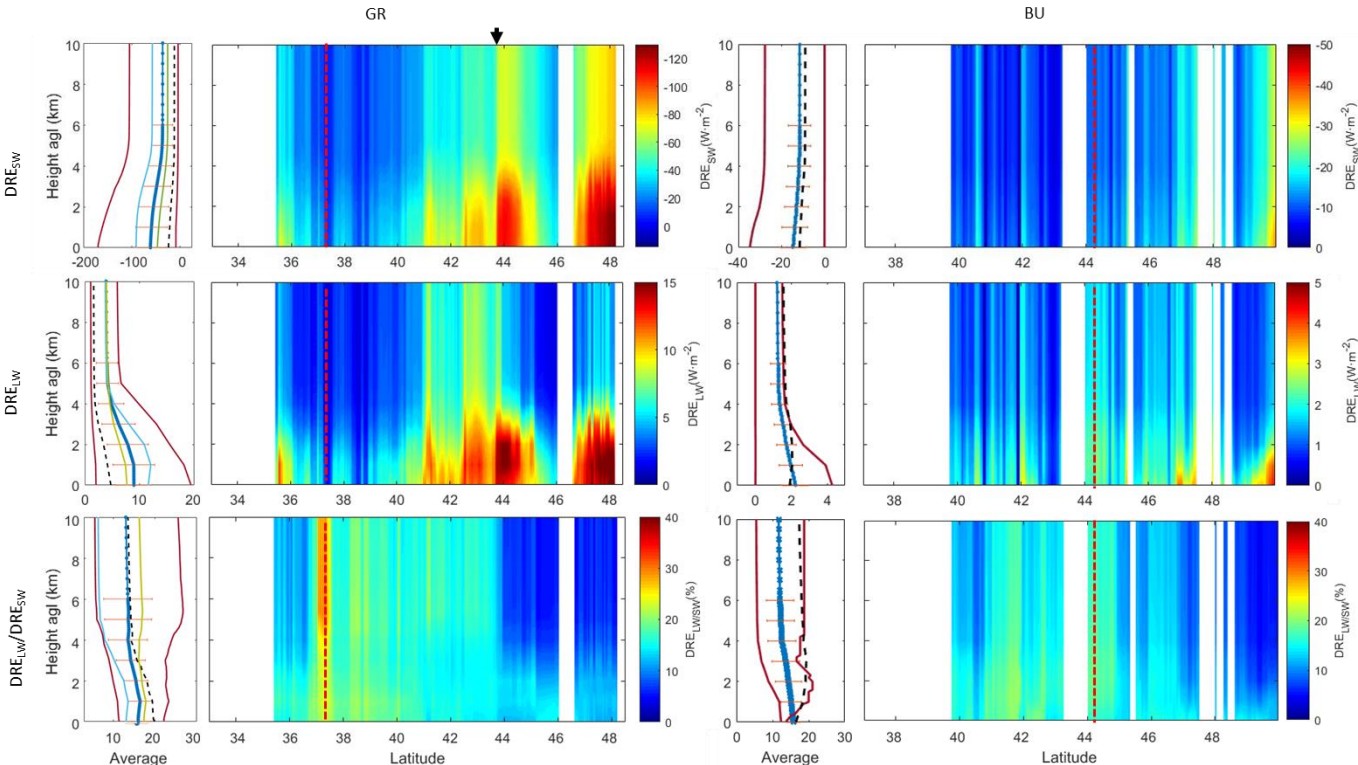

**Figure 7. Averaged profiles (blue lines) and standard deviation (orange error bars) and curtain plots of the DRE$_{SW}$, DRE$_{LW}$ and DRE$_{SW}$/DRE$_{LW}$ (from top to bottom) profiles obtained along the CALIPSO track for GR (left) and BU (right). Note the different scales. The red solid lines show the profiles corresponding to minimum and maximum AOD and the dashed black lines represents the profiles above the ground-based sites. For GR, the green and light blue lines show the averaged profiles for land and ocean respectively. The dashed red vertical lines represent the location of the EARLINET stations. The black downward arrow indicates the latitude where the surface shifts from land to ocean.**

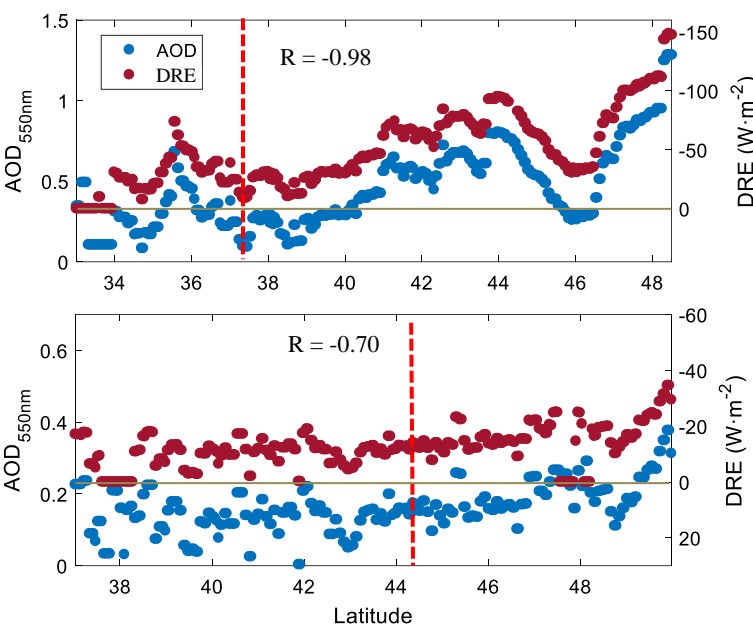

**Figure 8. AOD$_{550}$ from MODIS(blue dots) and DRE$_{BOA}$ from GAME (red dots) along the CALIPSO track for GR (top) and BU (bottom). The dashed red vertical lines represent the location of the EARLINET stations. The correlation coefficient between the DRE$_{BOA}$ and the AOD$_{550}$, R, is reported in each plot.**

The FE values are obtained along the CALIPSO tracks (Figure 9). On average, the FE at the BOA is much larger in absolute terms for GR. The obtained values range between -80.3 and -63.0 W·m$^{-2}$ for BU and -119.1 and -79.3 W·m$^{-2}$ for GR. For the TOA, similar average values are obtained at both stations, ranging between -79.5 and -36.8 W·m$^{-2}$ at GR and -69.4 and 43.3 W·m$^{-2}$ at BU. The GR case presents similar values to those observed in previous studies (Gómez-Amo et al., 2011; Peris-Ferrús et

al., 2017) whereas the retrieved FE for BU at the BOA is usually lower, even though it is necessary to consider the differences in SZA. The differences between the FE observed at GR and BU are related to the different aerosol properties observed in the AERONET data, but mostly related to the different SZA and the surface albedo (Gómez-Amo et al., 2011). A quite strong increase is observed in the GR track for latitudes higher than 44°, in coincidence with the decrease of the surface albedo and in agreement

with what we observed for the DRE. Average values above ocean are equal to -119.8 W·m$^{-2}$ at the BOA





and -81.7 W·m⁻² at the TOA, whereas over land the FE is much weaker due to the surface albedo, with FE being equal to -96.0 W·m⁻² at the BOA and -52.3 W·m⁻² at the TOA.

The track-averaged values obtained at the BOA and TOA are summarized in Table 3. The DRE (in absolute terms) is much larger in GR for both the SW and LW component. For this case, the aerosol

load is much higher as indicated by the averaged AOD$_{550}$ values along the track (0.52 for GR vs 0.16 for BU). In the case of the SW, a cooling of the surface is observed in both cases, with the averaged DRE$_{BOA}$=-66.7±35.7 and -14.6±5.2 W·m⁻² for GR and BU respectively. At the TOA, the DRE obtained is equal to -39.7±23.6 and -11.7±5.2 W·m⁻² for GR and BU respectively. For the LW fraction, positive values are obtained both at the BOA and the TOA, indicating a heating that partly counterbalances the

cooling of the SW spectral range. In general, the LW DRE fraction represents a 10-15% of the SW, with larger values near the surface. Overall, this fraction is in agreement with the values obtained in previous studies for the European region ranging between 9% and 26% (Perrone and Bergamo, 2011; di Sarra et al., 2011; Sicard et al., 2014b, 2014a ; Lolli et al., 2018; Meloni et al., 2018).

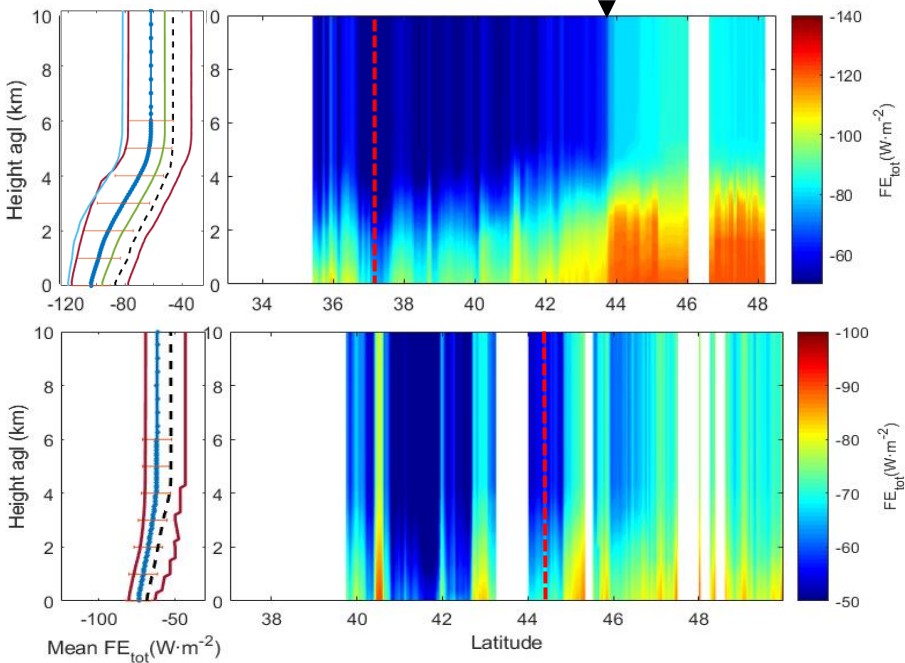

**Figure 9. Averaged profiles and standard deviation (error bars) and curtain plots of the FE along the CALIPSO track for GR (top) and BU (bottom) cases. The dashed red vertical lines represent the location of the EARLINET stations. The red solid lines show the profiles corresponding to minimum and maximum AOD and the dashed black lines represents the profiles above the**



**ground-based sites. For GR, the green and light blue lines show the averaged profiles for land and ocean respectively. The black downward arrow indicates the latitude where the surface shifts from land to ocean.**

**Table 3. DRE, FE and AOD average values ± standard deviation along the CALIPSO tracks for GR and BU cases.**

|  | GR | | BU | |
|---|---|---|---|---|
|  | **BOA** | **TOA** | **BOA** | **TOA** |
| **$DRE_{SW}$ (W·m$^{-2}$)** | -66.7 ±35.7 | -39.7±23.6 | -14.6± 6.3 | -11.7± 5.2 |
| **$DRE_{LW}$ (W·m$^{-2}$)** | 9.5±3.9 | 3.9±1.8 | 2.2 ±0.8 | 1.2 ±0.4 |
| **DRE (W·m$^{-2}$)** | -52.9±36.3 | -33.5±25.4 | -12.4 ±5.5 | -10.5± 5.0 |
| **$DRE_{SW}/DRE_{LW}$ (%)** | 16.0±2.9 | 13±5.6 | 15.8±2.1 | 11.7±3.7 |
| **FE (W·m$^{-2}$)** | -104.9±13.4 | -63.6±15.9 | -74.1 ±87 | -61.9± 9.5 |
| **$AOD_{550}$** | 0.52±0.25 | | 0.16± 0.06 | |

The SW vertical distribution of the HR is highly correlated with the $\alpha_{aer}$ values as well, when comparing Figure 3 and Figure 10, where larger HR values are observed in those layers with higher extinction. On average, we can see that the HR is maximum in the layer between 1 and 5 km for the GR case whereas it is very homogeneous below 4 km in the BU case and is in agreement with the $\alpha_{aer}$ profiles in Figure 3. The HR values are much lower in the BU case, since the aerosol load is low compared to the GR event. The averaged profiles obtained at GR reach up to 0.72 K·day$^{-1}$, whereas at BU they are lower by a factor of 10. The maximum value observed for the BU case is 0.25 K·day$^{-1}$ for altitudes around 1.5 km at high latitudes, whereas for GR values as large as 3.7 K·day$^{-1}$ at altitudes between 3.5 and 4 km are found from 44°N onwards. For the LW, the mineral dust effect is much lower and of opposite sign to that of the SW, as it occurs with the DRE values. In this case, the cooling rate values are as low as -0.18 K·day$^{-1}$ for GR and -0.02 K·day$^{-1}$ for BU on average. The maximum cooling rates are -0.52 K·day$^{-1}$ for GR and -0.35 K·day$^{-1}$ for BU. A slight heating can be observed near the surface, below the dust layers, especially in the GR event in the last part of the track (latitude > 42°). This result (negative LW HR values in the dust layer and positive below it) has been reported by Huang et al. (2009) and Wang et al. (2013), among others, and reflects the heating of the surface layers due to absorption of the LW radiation emitted by the dust layer above.



The cooling of the dust layers in the LW slightly counterbalances the heating due to the absorption of SW radiation. Net HR values on the averaged profiles reach a maximum of 0.63 K·day$^{-1}$ and 0.05 for GR and BU respectively, which are 0.09 and 0.02 K·day$^{-1}$ lower than the values in the SW (a decrease of 7 and 20% respectively). The averaged HR LW-to-SW ratios are similar in both cases, being 25% for GR and 28% for BU. In general terms, the HR values and the ratios obtained here are in quite good agreement with those observed by Huang et al., (2009) and Lemaître et al., (2010) considering that the aerosol load is much lower in our cases. LW and SW values for the HR are anticorrelated, with the strongest heating in the SW occurring in the same layers where the LW cooling is stronger and the aerosol load is higher.

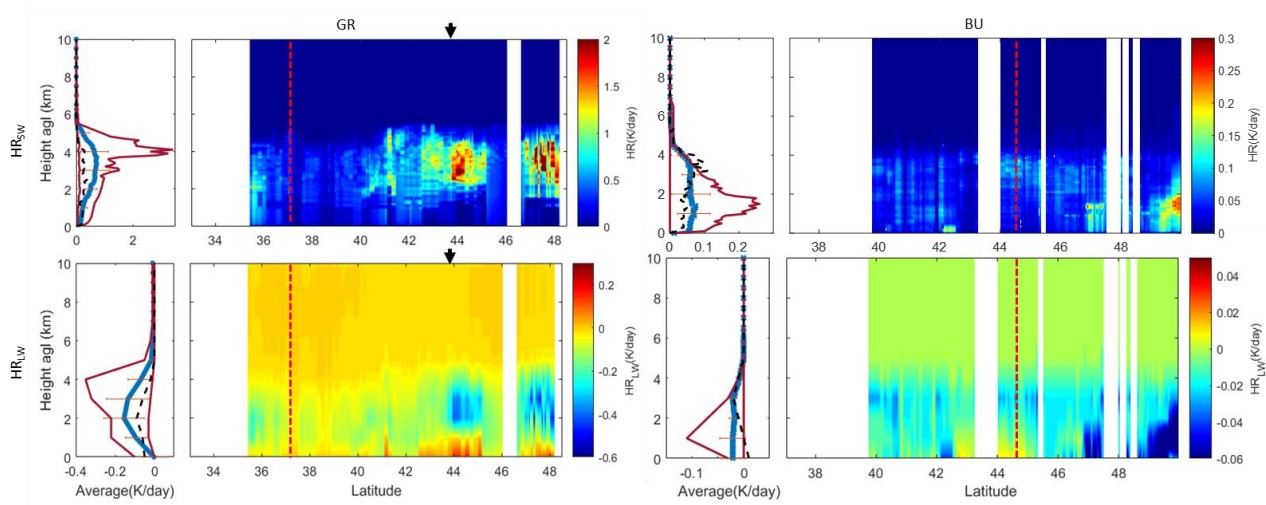

**Figure 10. Averaged profiles and standard deviation (error bars) and curtain plots of the HR profiles along the CALIPSO track for the SW (top) and LW (bottom) in the case of GR (left) and BU (right). The dashed red vertical lines represent the location of the EARLINET stations. The red solid lines show the profiles corresponding to minimum and maximum AOD and the dashed black lines represents the profiles above the ground-based sites. For GR, the black downward arrow indicates the latitude where the surface shifts from land to ocean.**



## 6.  Summary and conclusions

Two mineral dust events affecting Europe and detected by both CALIPSO and the EARLINET ground-based lidar sites of Granada and Bucharest are analyzed in order to examine the feasibility to retrieve 2-D mineral dust radiative effects using satellite data and the GAME radiative transfer model. The first dust event is a strong event affecting Western Europe on 7 April 2011 (GR event), whilst the second corresponds to a weaker event affecting Eastern Europe on 4 July 2012 (BU event).

The retrieval of radiative properties with GAME using an aerosol parameterization (the so-called PAR1) using CALIPSO vertical information with the AOD from MODIS and dust microphysical properties from AERONET is performed. A second aerosol parameterization from combined ground-based lidar and sun-photometer data (PAR2), which is already well established in the literature, is also applied in the present study to the EARLINET sites of Granada and Bucharest for the retrieval of radiative properties profiles. The aerosol direct radiative effects obtained with both parameterizations are intercompared for simultaneous and collocated CALIPSO and EARLINET lidar profiles in order to validate the PAR1, obtaining quite good agreement. Results indicate that the vertical distribution of the mineral dust provided by CALIPSO combined with MODIS AOD provides similar results to those obtained with the ground-based stations in spite of the higher noise in the CALIPSO lidar profiles.

Once the satellite-based methodology is validated, the 2-D DRE in both the SW and LW is calculated along CALIPSO tracks for both cases. In general terms, a strong correlation is observed between the absolute value of the DRE at the surface and MODIS AOD values, with correlation coefficients of -0.98 for GR and -0.70 for BU. Consequently, the DRE and HR are larger in the case of GR, where the average AOD along the orbit is equal to 0.52, much larger than the 0.16 value obtained for BU. DRE values at the surface reach up to -169.8 $W \cdot m^{-2}$ (SW) and 20 $W \cdot m^{-2}$ (LW) in the GR case, and -38.0 $W \cdot m^{-2}$ (SW) and 4.0 $W \cdot m^{-2}$ (LW) in the BU case, and are in good agreement with previous studies. As expected, the LW contribution is of opposite sign to the SW and much lower (less than 25%). The effect of the LW counterweights the cooling in the SW range, and the averaged net DRE values at the BOA and TOA are -52.9 and -33.5 $W \cdot m^{-2}$ for GR and -12.4 and -10.5 $W \cdot m^{-2}$ for BU, which trans-



lates into a decrease of the DRE ranging between 10 and 20% compared to the SW values. For the FE, the obtained values range between -80.3 and -63.0 W·m$^{-2}$ for BU and -119.1 and -79.3 W·m$^{-2}$ for GR at the BOA. Different values are obtained for both cases due to differences in the incoming radiation, dust properties and the surface albedo. The strong influence of the surface albedo in the SW radiative properties is evident in the GR case, where the shift from land to surface has a strong impact on the DRE and the FE values. Regarding the HR, LW and SW values are also anticorrelated, with positive values observed for the SW and negative values for the LW. A stronger effect is observed in those layers containing higher aerosol load (between 1 and 5 km for GR and below 4 km for BU). The averaged profiles obtained at GR reach up to 0.72 K·day$^{-1}$ and 0.07 K·day$^{-1}$ at BU for the SW, whereas for the LW values are as low as -0.18 K·day$^{-1}$ for GR and -0.02 K·day$^{-1}$ for BU. Net HR are a 7 and 20% lower than the SW values for GR and BU respectively.

The presented methodology based on satellite data allows us to extend the analysis of radiative properties with GAME from one to two dimensions by greatly increasing spatial and temporal coverage. In the present study, AERONET data are used to define the aerosol microphysical properties, which is currently not a limitation because of the large spatial extent of the network in regions such as Europe and North America. Nonetheless, the use of accurate microphysical properties from satellite data would greatly increase the use of this methodology without the need for ground-based instrumentation. Although this work is focused on the long-range transport of mineral dust, our methodology is easily applicable to the long-range transport of other aerosol types such as fire smoke or volcanic ash.

**Author contribution**

MJGM, MS and NP designed the study and wrote the manuscript with contributions from all authors. MJGM, NP, RB, JABA and DN provided data and performed data analysis. MJGM performed the model simulations with the contribution of RB. All authors have given approval to the final version of the manuscript.





## Acknowledgments

This work was supported by the European Union through H2020 programme (ACTRIS-2, grant 654109; ECARS, grant 602014; EUNADICS-AV, grant 723986; GRASP-ACE, grant 778349). The Barcelona team acknowledges the Spanish Ministry of Economy and Competitivity (project TEC2015-63832-P) and EFRD (European Fund for Regional Development); the Spanish Ministry of Science, Innovation and Universities (project CGL2017-90884-REDT); and the Unidad de Excelencia Maria de Maeztu (project MDM-2016-0600) financed by the Spanish Agencia Estatal de Investigación. The authors would also like to acknowledge the CALIPSO mission scientists and associated NASA personnel for the production of the data used in this research.

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
