# Peer review of "2-D mineral dust radiative effect calculations from CALIPSO observations over Europe"

_Atmospheric Chemistry and Physics, 2019_

## Referee Comment (RC1) · Anonymous Referee #2 · 10 Jul 2019

General comments The paper focuses on analyzing the dust radiative effects in SW and LW spectral ranges combining satellite data (CALIPSO, MODIS, etc) and radiative transfer modelling (GAME). The authors use also ground-based aerosol profiles, taking advantage of the simultaneous and quasi-collocated EARLINET stations, for validation. Despite this, the most interesting part of this work i the application to CALIPSO profiles. The radiative forcing and heating rates obtained show good agreement when comparing with the obtained using ground-based aerosol profiles and the same GAME RTM. I think the paper of interest for aerosol research and modelling community due to the great potential of CALIPSO data to be operationally used in model assimilation. Therefore, the argument and application of this paper is solid and then suitable for pub-

lication in ACP. However, several minor issues should be addressed before the paper is published.

Specific comments Abstract L8: What does the forcing efficiency variability depend on? solar zenith angle, surface type and albedo? Pag 11. L9: At Granada, we cannot consider EARLINET and CALIPSO as collocated measurements, since the distance between both is more than 100km. What implications could this distance have in the operational assimilation of CALIPSO data by the modelling community? Pag 12.L22: I do not understand why the authors describe so detailed the meteorological variables, even using a big plot, if then, this variables are not taken into account neither realted to the radiative forcing analysis. Pag 12.L25: The sentence "..general, even though a stratospheric O3 intrusion is affecting 25 the GR orbit increasing the O3 concentration in some regions." is difficult to understand can you rewrite it, pleae?

Pag 13.L14: What contrained by MODIS AOD really means? Is not clear if you are using the alfa profile by CALIPSO or you are inverting the Lidar raw data using MODIS AOD? Pag 15.L10: Please, remove the second "increasing" in the sentence

Figure 1 is confusing, can you use different colors for the EARLINET stations and for the CALIPSO track to better distinguish them? can you include more lat and long values to improve the visual appreciation of the distances? Figure 10. Caption: Please specify "Average heating rate profiles and standard deviation"...

---

## Referee Comment (RC2) · Anonymous Referee #1 · 10 Jul 2019

The manuscript by M.J. Granados-Munoz et al. 2019 aims to retrieve the radiative effects together with the heating rate along the atmospheric column during a dust outbreak. The work well fits within the journal scope and it is generally well written. However some points should be clarified by the authors before publication

First of all, "effect" and "forcing" are not interchangeable words and should be correctly employed. "Forcing" is referred to computations with respect to the "pre-industrial" era. For this reason, the term "effect" should be used in this manuscript.

It cannot be inferred from the text how the authors retrieved both the heating rate and dust radiative effects. It looks like that the authors assume the whole extinction

profile as dust, while surely in the boundary layer dust (if present) will blend with local background aerosols, i.e. urban aerosol. To calculate then the DRE, it should be considered the extinction profile as mixed, with respect to the altitude. I would say that g, SSA should be different in the boundary layer (polluted dust) with respect to the free troposphere (pure dust).Computations using range independent g and SSA values (as those retrieved from AERONET) will produce different results than considering two range dependent aerosol layers (with different SSA and g values).

In the manuscript it is stressed that two different parameterizations are used, i.e. one for EARLINET and one for CALIPSO. However, the different parameterizations reduce just to the different retrieval of the atmospheric extinction profile by the two different lidar system. Some section titles are then misleading and it worths to better explain the so-called "different parameterization".

The radiative transfer computations are computed up to 20 km. At this altitude, the effects of stratospheric ozone is not taken into consideration. Moreover, how the radiative calculations are carried out? Atmosphere with aerosol minus pristine atmosphere? Please advise.

Specific comments can be found in the attached file.

Please also note the supplement to this comment:
https://www.atmos-chem-phys-discuss.net/acp-2019-440/acp-2019-440-RC2-supplement.pdf

**Supplement:**

[revised manuscript text omitted]

---

## Author Comment (AC1) · 8 Aug 2019

**General comments**

**The paper focuses on analyzing the dust radiative effects in SW and LW spectral ranges combining satellite data (CALIPSO, MODIS, etc) and radiative transfer modelling (GAME). The authors use also ground-based aerosol profiles, taking advantage of the simultaneous and quasi-collocated EARLINET stations, for validation. Despite this, the most interesting part of this work I the application to CALIPSO profiles. The radiative forcing and heating rates obtained show good agreement when comparing with the obtained using ground-based aerosol profiles and the same GAME RTM. I think the paper of interest for aerosol research and modelling community due to the great potential of CALIPSO data to be operationally used in model assimilation. Therefore, the argument and application of this paper is solid and then suitable for publication in ACP. However, several minor issues should be addressed before the paper is published.**

We would like to thank the reviewer for his/her valuable comments and remarks. Responses to the specific comments are provided next:

**Specific comments**

**Abstract L8: What does the forcing efficiency variability depend on? solar zenith angle, surface type and albedo?**

It depends mostly on the SZA and the surface albedo, together with the aerosol vertical distribution. The SSA and g are also affecting the FE, but in our case these aerosol properties remain constant along the orbit tracks.

**Pag 11. L9: At Granada, we cannot consider EARLINET and CALIPSO as collocated measurements, since the distance between both is more than 100km. What implications could this distance have in the operational assimilation of CALIPSO data by the modelling community?**

For this case the distance is more than 100 km, but, as indicated in the text, the severity of the dust event suggested similar dust characteristics for CALIPSO and EARLINET. In general and specifically for operational assimilation purposes the threshold of 100 km should be kept. Cases as the one presented here are exceptional and special consideration would be needed for assimilation in models.

**Pag 12.L22: I do not understand why the authors describe so detailed the meteorological variables, even using a big plot, if then, this variables are not taken into account neither related to the radiative forcing analysis.**

The meteorological data are taken into account in order to retrieve the radiative fluxes which are necessary to obtain the DRE. They are used as input in the GAME radiative transfer model. This sentence has been added to the manuscript:

"Meteorological profiles, including vertically-resolved temperature (T), relative humidity (RH) and $O_3$ concentration, are necessary for the calculation of the radiative fluxes in GAME."

**Pag 12.L25: The sentence "..general, even though a stratospheric O3 intrusion is affecting 25 the GR orbit increasing the O3 concentration in some regions." is difficult to understand can you rewrite it, pleae?**

Done. The sentence now reads

P12, l27: "Larger T values and $O_3$ concentration are observed for the BU case in general. However, $O_3$ values larger than 0.07 ppmv are observed in the GR orbit related to a stratospheric intrusion."

**Pag 13.L14: What contrained by MODIS AOD really means? Is not clear if you are using the alfa profile by CALIPSO or you are inverting the Lidar raw data using MODIS AOD?**

We use the extinction profile provided by CALIPSO multiplied by the ratio of MODIS AOD to CALIPSO AOD. The information is now included in the manuscript:

P12, l11: "The first dataset (hereafter denoted as DAT1), presented in the current study, uses satellite-based extinction profiles from CALIPSO constrained by MODIS AOD, meaning that CALIPSO extinction profiles are normalized so that the integral of the extinction profile matches MODIS AOD."

**Pag 15.L10: Please, remove the second "increasing" in the sentence**

We think that the reviewer refers to Page 14, line 20. The second "increasing" is removed.

**Figure 1 is confusing, can you use different colors for the EARLINET stations and for the CALIPSO track to better distinguish them? can you include more lat and long values to improve the visual appreciation of the distances?**

The colors in the figure have been modified. By increasing the spatial coverage in the figure it is difficult to observe the variations in the AOD along the orbit, thus we would rather maintain the current latitude and longitude ranges.

**Figure 10. Caption: Please specify "Average heating rate profiles and standard deviation"...**

It is specified afterwards in the caption: "Averaged profiles and standard deviation (error bars) and curtain plots of the HR profiles"

---

## Author Comment (AC2) · 8 Aug 2019

**The manuscript by M.J. Granados-Munoz et al. 2019 aims to retrieve the radiative effects together with the heating rate along the atmospheric column during a dust outbreak. The work well fits within the journal scope and it is generally well written. However some points should be clarified by the authors before publication First of all, "effect" and "forcing" are not interchangeable words and should be correctly employed. "Forcing" is referred to computations with respect to the "pre-industrial" era. For this reason, the term "effect" should be used in this manuscript.**

First of all, we would like to thank the reviewer for his/her valuable comments and remarks. We agree that the term "forcing" is misused in the manuscript and it has been replaced by the term "effect" throughout the text.

**It cannot be inferred from the text how the authors retrieved both the heating rate and dust radiative effects. It looks like that the authors assume the whole extinction profile as dust, while surely in the boundary layer dust (if present) will blend with local background aerosols, i.e. urban aerosol. To calculate then the DRE, it should be considered the extinction profile as mixed, with respect to the altitude. I would say that g, SSA should be different in the boundary layer (polluted dust) with respect to the free troposphere (pure dust). Computations using range independent g and SSA values (as those retrieved from AERONET) will produce different results than considering two range dependent aerosol layers (with different SSA and g values).**

The reviewer is right, the extinction profile is considered to be caused solely by being mineral dust. This information is included in the manuscript in Page 12, lines 13-15: "The total extinction profiles and AOD values considered in the study are assumed to be due to the mineral dust particles, even though some other aerosol types might be present in the mixture."

According to CALIPSO typing (see figure below), the majority of the profiles correspond to mineral dust or polluted dust, therefore the uncertainty associated with this assumption is expected to be low. We agree with the reviewer that results would be more accurate with vertically resolved SSA and g values, but unfortunately these data are not available from any collocated and simultaneous satellite measurement. Anyhow, the uncertainty due to the assumptions that the whole extinction is due to mineral dust and that SSA and g are constant is expected to be almost negligible, since the main drive of the DRE is the AOD (e.g. Granados-Muñoz et al., 2019). The text has been modified considering this information.

P14, l19-21: "*The assumption of constant SSA, g and PSD values is not exempt of uncertainty, but this is expected to be almost negligible, since the main drive of the DRE is the AOD (e.g. Granados-Muñoz et al., 2019).*"

[Figure]

N|A = not applicable   1 = marine   2 = dust   3 = polluted continental/smoke   4 = clean continental   5 = polluted dust   6 = elevated smoke   7 = dusty marine
8 = PSC aerosol   9 = volcanic ash   10 = sulfate/other

Figure. CALIPSO aerosol typing provided on 7 April 2011 (GR, left) and 4 July 2012 (BU, right).

**In the manuscript it is stressed that two different parameterizations are used, i.e. one for EARLINET and one for CALIPSO. However, the different parameterizations reduce just to the different retrieval of the atmospheric extinction profile by the two different lidar system. Some section titles are then misleading and it worths to better explain the so-called "different parameterization".**

Exactly, the only difference in the parameterizations is the use of the extinction profiles, one for the ground system and one for the satellite measurements. We agree that the term parameterization may be misleading and it has been substituted by the term "dataset" where appropriate.

**The radiative transfer computations are computed up to 20 km. At this altitude, the effects of stratospheric ozone is not taken into consideration. Moreover, how the radiative calculations are carried out? Atmosphere with aerosol minus pristine atmosphere? Please advise. Specific comments can be found in the attached file.**

Yes, the aerosol forcing is calculated as:

$$DRE = (F^w_{down} - F^w_{up}) - (F^o_{down} - F^o_{up})$$

where $F^w$ and $F^o$ are the radiative fluxes with and without aerosols, respectively. This information is now included in the manuscript.

Because we are interested in the aerosol radiative effect, the effect of stratospheric ozone is not so relevant in the presented results. Results obtained when recalculating the profiles in the model up to 80 km and considering the contribution of stratospheric ozone are basically the same.

**Please also note the supplement to this comment: https://www.atmos-chem-phys-discuss.net/acp-2019-440/acp-2019-440-RC2supplement.pdf**

Corrections indicated by the reviewer have been implemented in the new version of the manuscript. Please, find below the responses to the questions arose by the reviewer throughout the text in the supplementary material:

**P9, l11: Are the extinction profiles and met data adapted to GAME model vertical levels?**

Yes, the resolution of the extinction profiles and meteorological data is degraded to the resolution of the model.

**P9, l12: 20 km is not enough, as ozone and other variables play a role up to 90km. How the results change if integrated up to 90km?**

The mineral dust profiles do not vary when considering an altitude range in the model up to 90 km, since the analysis performed here focuses only on the aerosol effect.

**P12, l7-20: I am not sure to get the point, but the difference between the two parameterization is merely depending on the atmospheric extinction coefficient? I would then rather make a sensitivity study of GAME with respect of the extinction coeff**

The reviewer is right, the difference is mainly the extinction profiles. In one case, extinction profiles from the ground-based lidar system constrained by AERONET AOD are used, whereas in the other case the satellite-based extinction profiles constrained by MODIS AOD are used. The idea is to evaluate if the differences in the vertical distribution and the expected noise in the CALIPSO profiles have a strong impact in the results of the DRE. A sensitivity study of the extinction values is out of the scope of this study. It was already performed in Granados-Muñoz et al. (2019) and it was obtained that variations in the AOD of 0.05 led to variations in the DRE of up to 20% at the surface.

**P13, l3: Which product? please see https://journals.ametsoc.org/doi/abs/10.1175/JAMC-D-16-0262.1**

We thank the reviewer for this reference. Information about the product used here is included in Page 6, line 22: "For the spectral surface albedo, the MODIS MCD43C3 Version 6 Albedo Model data set is used (Schaaf and Wang, 2015), providing values at 7 different wavelengths between 0.670 and 2.155 µm."

**P14, l1: MODIS AOD retrieval is performed for the whole bandwidth instead for a single wavelength as for lidar. Is this difference taken into consideration?**

The MODIS product used in our work (see Section 2.1.2) is the AOD at the wavelength of 550 nm taken from the MODIS standard C6 MYD04_L2 product. The raw signal comes from band #4 of MODIS which has a bandwidth comprised between 545 and 565 nm. CALIOP visible wavelength is at 532 nm and the bandwidth is < 1 nm. We believe that the different bandwidths have an effect mostly on the background noise of each sensor.

**P15, l5: Be careful because Level 1.5 data have pre-field calibration applied, however the calibration can change during the deployment (usually a linear rate due to slow deposition on the sensor head lenses), hence, the need for a post-field calibration. This means that Level 1.5 may show a large bias.**

In the case of the data used here both pre- and post-calibration have been applied, since corresponding Level 2 data are available for that period. The data used here did not reach Level 2.0 because they do not fulfill the AOD (AOD>0.4) or the SZA (SZA>50 deg) criteria.

**P18, l10: those albedo values seem to be very low, e.g. over water. For GR it could be, but not for BA**

It is necessary to take into consideration that they are provided for 470 nm and values at this wavelength are usually quite low. For longer wavelengths (e.g. 1020 nm) larger values are obtained (0.32 for BU and 0.27 for GR).

**P19, l1: how the overlap function is considered for the EARLINET system? How is extrapolated the extinction value from lowest point to surface?**

The extinction in the incomplete overlap region is assumed to be constant and equal to the value of the lowest point of the extinction profile. This information is now included in the text (P16, l7-9): "*The incomplete overlap for the EARLINET systems has not been corrected here; the $\alpha_{aer}$ values in the affected region are assumed to be constant and equal to the closest to the surface valid value in the profile*"